# Probability of sepsis after infection consultations in primary care in the United Kingdom in 2002–2017: Population-based cohort study and decision analytic model

**Martin C. Gulliford**[1,2]*, **Judith Charlton**[1], **Joanne R. Winter**[1], **Xiaohui Sun**[1], **Emma Rezel-Potts**[1,2], **Catey Bunce**[1,2], **Robin Fox**[3], **Paul Little**[4], **Alastair D. Hay**[5], **Michael V. Moore**[4], **Mark Ashworth**[1], **SafeAB Study Group**[¶]

1 School of Population Health and Environmental Sciences, King's College London, London, United Kingdom, 2 NIHR Biomedical Research Centre at Guy's and St Thomas' Hospitals London, London, United Kingdom, 3 The Health Centre, Bicester, United Kingdom, 4 Primary Care Research Group, University of Southampton, Aldermoor Health Centre, Southampton, United Kingdom, 5 Centre for Academic Primary Care, Bristol Medical School, Population Health Sciences, University of Bristol, Bristol, United Kingdom

¶ Membership of the SafeAB Study Group is provided in the Acknowledgements.

* martin.gulliford@kcl.ac.uk

**Data Availability Statement:** Data cannot be shared publicly because they are analysed under licence. Permission for data access is through the

## Abstract

### Background

Efforts to reduce unnecessary antibiotic prescribing have coincided with increasing awareness of sepsis. We aimed to estimate the probability of sepsis following infection consultations in primary care when antibiotics were or were not prescribed.

### Methods and findings

We conducted a cohort study including all registered patients at 706 general practices in the United Kingdom Clinical Practice Research Datalink, with 66.2 million person-years of follow-up from 2002 to 2017. There were 35,244 first episodes of sepsis (17,886, 51%, female; median age 71 years, interquartile range 57–82 years). Consultations for respiratory tract infection (RTI), skin or urinary tract infection (UTI), and antibiotic prescriptions were exposures. A Bayesian decision tree was used to estimate the probability (95% uncertainty intervals [UIs]) of sepsis following an infection consultation. Age, gender, and frailty were evaluated as association modifiers. The probability of sepsis was lower if an antibiotic was prescribed, but the number of antibiotic prescriptions required to prevent one episode of sepsis (number needed to treat [NNT]) decreased with age. At 0–4 years old, the NNT was 29,773 (95% UI 18,458–71,091) in boys and 27,014 (16,739–65,709) in girls; over 85 years old, NNT was 262 (236–293) in men and 385 (352–421) in women. Frailty was associated with greater risk of sepsis and lower NNT. For severely frail patients aged 55–64 years, the NNT was 247 (156–459) in men and 343 (234–556) in women. At all ages, the probability of sepsis was greatest for UTI, followed by skin infection, followed by RTI. At 65–74 years, the NNT following RTI was 1,257 (1,112–1,434) in men and 2,278 (1,966–2,686) in women; the

CPRD Independent Scientific Advisory Committee (ISAC, contact via isac@mhra.gov.uk) for researchers who meet the criteria for access to confidential data. The data underlying the results presented in the study are available from the Clinical Practice Research Datalink (CPRD, cprdenquiries@mhra.gov.uk).

**Funding:** The study is funded by the National Institute for Health Research (NIHR) Health Services and Delivery Programme (16-116-46). The funder of the study had no role in study design, data collection, data analysis, data interpretation, or writing of the report.

**Competing interests:** The authors have declared that no competing interests exist.

**Abbreviations:** AMR, antimicrobial drug resistance; APC, admitted patient care; CPRD, Clinical Practice Research Datalink; HES, Hospital Episode Statistics; ISAC, Independent Scientific Advisory Committee; NEWS2, National Early Warning Score; NNT, number needed to treat; ONS, Office for National Statistics; RTI, respiratory tract infection; STROBE, Strengthening the Reporting of Observational Studies in Epidemiology; UI, uncertainty interval; UTI, urinary tract infection..

NNT following skin infection was 503 (398–646) in men and 784 (602–1,051) in women; following UTI, the NNT was 121 (102–145) in men and 284 (241–342) in women. NNT values were generally smaller for the period from 2014 to 2017, when sepsis was diagnosed more frequently. Lack of random allocation to antibiotic therapy might have biased estimates; patients may sometimes experience sepsis or receive antibiotic prescriptions without these being recorded in primary care; recording of sepsis has increased over the study period.

## Conclusions

These stratified estimates of risk help to identify groups in which antibiotic prescribing may be more safely reduced. Risks of sepsis and benefits of antibiotics are more substantial among older adults, persons with more advanced frailty, or following UTIs.

## Author summary

### Why was this study done?

- Sepsis is a severe reaction to an infection that may lead to life threatening damage to organ systems. Sepsis is an increasingly recognised concern for health professionals and patients in primary care.

- Inappropriate and unnecessary antibiotic prescribing is a widespread problem in primary care that may be contributing to antimicrobial resistance.

- This study aimed to estimate the probability of a patient developing sepsis after an infection consultation in primary care if antibiotics are or are not prescribed.

### What did the researchers do and find?

- We analysed the electronic health records of all registered patients at 706 general practices, with 66.2 million person-years of follow-up from 2002 to 2017 and 35,244 first episodes of sepsis.

- We found that the probability of sepsis was lower if an antibiotic was prescribed, but the number of antibiotic prescriptions required to prevent one episode of sepsis (number needed to treat [NNT]) decreased with age.

- Frailty was associated with greater risk of sepsis and lower NNT.

- At all ages, the probability of sepsis was greatest for urinary tract infection, followed by skin infection, followed by respiratory tract infection.

### What do these findings mean?

- These results show that risks of sepsis and benefits of antibiotics are more substantial among older adults, persons with more advanced frailty, or following urinary tract infections.

- Antibiotic use may be more safely reduced in groups with lower probability of sepsis.

- We caution that our results represent averages over diverse localities and years of study, and lack of random allocation to antibiotic therapy might have caused bias.

## Introduction

The threat of antimicrobial drug resistance (AMR) is attracting the concern of national governments and international organisations [1]. Antibiotic-resistant infections are increasing and are more often identified in primary care as well as hospital settings. In the UK, antibiotic prescribing in primary care accounts for more than three-quarters of all antibiotic use. Respiratory tract infections (RTIs) represent the most common reason for antibiotic treatment [2], with general practitioners prescribing antibiotics at about half of the consultations for 'self-limiting' RTIs, including common colds, acute cough and bronchitis, sore throat, otitis media, and rhinosinusitis [3], with little change over the last 2 decades [4,5]. The other main indications for antibiotic prescription include urinary tract infections (UTIs) and skin infections [2,6,7]. The UK government has developed a 5-year antimicrobial resistance strategy that identifies reducing unnecessary antibiotic prescribing and improving antibiotic selection as key elements of antimicrobial stewardship [8,9].

Reducing antibiotic use might potentially compromise patient safety by increasing the risk of serious bacterial infections following consultations for common infections [10]. The safety of reduced antibiotic prescribing is a major concern for both clinicians and patients [11]; parents may also be particularly concerned about safety issues, which are often an important motivation for seeking active treatment for children [12]. A systematic review of qualitative studies found that clinicians commonly prescribe antibiotics 'just in case' they might be needed [13]. Based on international comparisons, with both low [14] and high [15] antibiotic prescribing being observed across Europe without apparent risks to patient safety, it appears that a substantial reduction of antibiotic prescribing in primary care might be reasonable. However, only a few existing research studies directly address the safety outcomes of reduced antibiotic prescribing at consultations for common infections in primary care.

Strategies to reduce inappropriate use of antibiotics must ensure that antibiotics can be used when they are needed [16,17]. Bacterial infections are still of public health importance, and there has been growing recognition of the importance of sepsis, with more than 200,000 hospital admissions for sepsis each year in England and up to 59,000 deaths [18]. Early recognition and treatment of sepsis is being promoted by health services and professional organisations through assessment of risk for individual patients [19]. In the UK, a National Early Warning Score (NEWS2) based on six physiological parameters has been promoted to identify individual patients who may be at risk of sepsis [20]. However, this approach has also been criticised because early warning signs of sepsis are often nonspecific, and alerting systems may result in false-positive signals at many consultations [21].

Research is needed to provide quantitative estimates of risk that might provide clinicians and patients with evidence to inform antibiotic prescribing decisions. This study aimed to estimate the probability of sepsis if antibiotics were prescribed or not and to estimate the number of antibiotic prescriptions required to prevent one episode of sepsis. We estimated the probability of sepsis for groups of patients characterised by age, gender, and frailty as well as reason for consultation.

## Methods

### Ethics statement

Scientific and ethical approval of the protocol was given by the Clinical Practice Research Datalink (CPRD) Independent Scientific Advisory Committee (ISAC protocol 18-041R). The study was based on analysis of fully anonymised data, and individual consent was not required.

### Data source

We carried out a population-based cohort study in the UK CPRD GOLD database, employing data for 2002–2017. The CPRD GOLD (www.cprd.com) is one of the world's largest databases of primary care electronic health records, with participation of about 7% of UK family practices and with ongoing collection of anonymised data from 1990 [22]. CPRD GOLD is considered to be geographically and sociodemographically representative of the UK population [22]. The high quality of CPRD GOLD data has been confirmed in many studies [23]. The protocol for the study has been published (https://fundingawards.nihr.ac.uk/award/16/116/46). Descriptive data for antibiotic prescribing and general practice–level associations have been reported previously [24]. This study is reported as per the Strengthening the Reporting of Observational Studies in Epidemiology (STROBE) guideline (S1 STROBE Checklist).

### Sepsis events

We ascertained sepsis events from the entire registered population of CPRD GOLD because these are generally rare events. Incident cases of sepsis were obtained from CPRD GOLD for the years 2002–2017, with person-years at risk providing the denominator. The start of the patient record was the latest of 1 year after the patient's current registration date, the date the general practice began contributing up-to-standard data to CPRD GOLD, or 1 January 2002. The end of the patient's record was defined as the earliest of the end of registration, the patient's death date, or 31 December 2017. The mean duration of follow-up was 6.9 years. Sepsis events were evaluated using Read codes recorded into patients' clinical and referral records [24]. There were 77 Read codes for sepsis and septicaemia, but the four most frequent codes accounted for 92% of events including 'Sepsis' (two codes), 'Septicaemia', and 'Urosepsis' (S1 Table). We included incident first events in further analyses; recurrent events in the same patient were not evaluated further because it may not always be possible to distinguish new occurrences from reference to ongoing or previous problems in electronic health records.

For each sepsis event, we evaluated whether a consultation for a common infection was recorded within the preceding 30 days. We employed a 30-day time window with the intention of capturing data for acute infections and their short-term outcomes. We identified consultations for RTIs (including upper and lower RTIs), skin infections, and UTIs (including cystitis and uncomplicated UTIs only) because these are the most important groups of conditions for which antibiotics are prescribed in primary care [25] (S2 Table). We evaluated Read codes in patients' clinical and referral records in order to identify consultations associated with common infections. We also evaluated whether an antibiotic prescription was issued during the 30 days preceding a sepsis event, either on the same date as an infection consultation or on a different date [24,25] (S3 Table).

### Selection of sample for antibiotic prescribing analysis

We estimated infection consultation rates and the proportion of consultations with antibiotics prescribed from a sample of patients registered with CPRD GOLD. This was because it is not

feasible to download and analyse data for the millions of records represented by all infection consultations and antibiotic prescriptions over 16 years [24]. A random sample of patients was drawn from the list of all registered patients, stratifying by year between 2002 and 2017 and by family practice. The 'sample' command in the R programme was employed to provide a computer-generated random sequence. In each year of study, a sample of 10 patients was taken for each gender and age group using 5-year age groups up to a maximum of 104 years. Each sampled patient contributed data in multiple years of follow-up. There was a total sample of 671,830 individual patients registered at a total of 706 family practices who contributed person-years between 2002 and 2017. The sampling design enabled estimation of all age-specific rates with similar precision, and age-standardisation provided weightings across age groups. Data for antibiotic prescribing in this sample have been reported previously [24] (S4 Table).

For each patient in the antibiotic prescribing sample, we calculated the person-years at risk between the start and end of the patient's record. Person-years was grouped by gender, age group, and comorbidity. Age groups were from 0 to 4, 5 to 9, and 10 to 14 years and then 10-year age groups up to 85 years and over. Infection consultations were evaluated using Read codes as outlined above. Antibiotic prescriptions were evaluated using product codes for antibiotics listed in section 5.1 of the British National Formulary, excluding methenamine and drugs for tuberculosis and leprosy. Different antibiotic classes and antibiotic doses were not considered further in this analysis. Multiple antibiotic prescription records on the same day were considered as a single antibiotic prescription.

## Evaluation of frailty

We used Clegg's e-Frailty Index to evaluate frailty level [26]. The e-Frailty Index includes 36 deficits, which are evaluated as present or absent based on Read-coded electronic health records. Patients were classified as being 'nonfrail' or having 'mild', 'moderate', or 'severe' frailty based on the number of deficits recorded. We evaluated frailty for each patient in each calendar year of study [27] in order to provide a frailty estimate for the index year of each sepsis episode. We also estimated consultation rates and antibiotic prescribing proportions by frailty category for the antibiotic prescribing sample. As full electronic health record data were not available for the entire CPRD GOLD denominator, we allocated person-years to frailty categories using the proportion in each frailty category that we observed in the antibiotic prescribing sample. Although the concept of frailty may be applied at any age, frailty was only evaluated from 55 years and older because most patients under the age of 55 years were classed as nonfrail or as having only mild frailty (S5 Table).

## Decision tree

In order to evaluate the probability of sepsis following an infection consultation in primary care, we constructed a decision tree (Fig 1) [28]. An individual developing an infection may decide to consult their general practice or not; if they consult, they may be prescribed antibiotics or not; subsequently, they may develop sepsis or not. We used estimates from CPRD data analysis to populate the decision tree with empirical estimates for probabilities as outlined in Table 1. We used Bayes' theorem to estimate the probability of sepsis following an infection consultation if antibiotics were prescribed or if antibiotics were not prescribed. We estimated the 'number needed to treat' (NNT), the number of antibiotic prescriptions required to prevent one sepsis event, as the reciprocal of the difference in probability of sepsis with and without antibiotics. We obtained central estimates and 95% uncertainty intervals from 10,000 random draws from the beta distribution [29]. All estimates were stratified by gender and

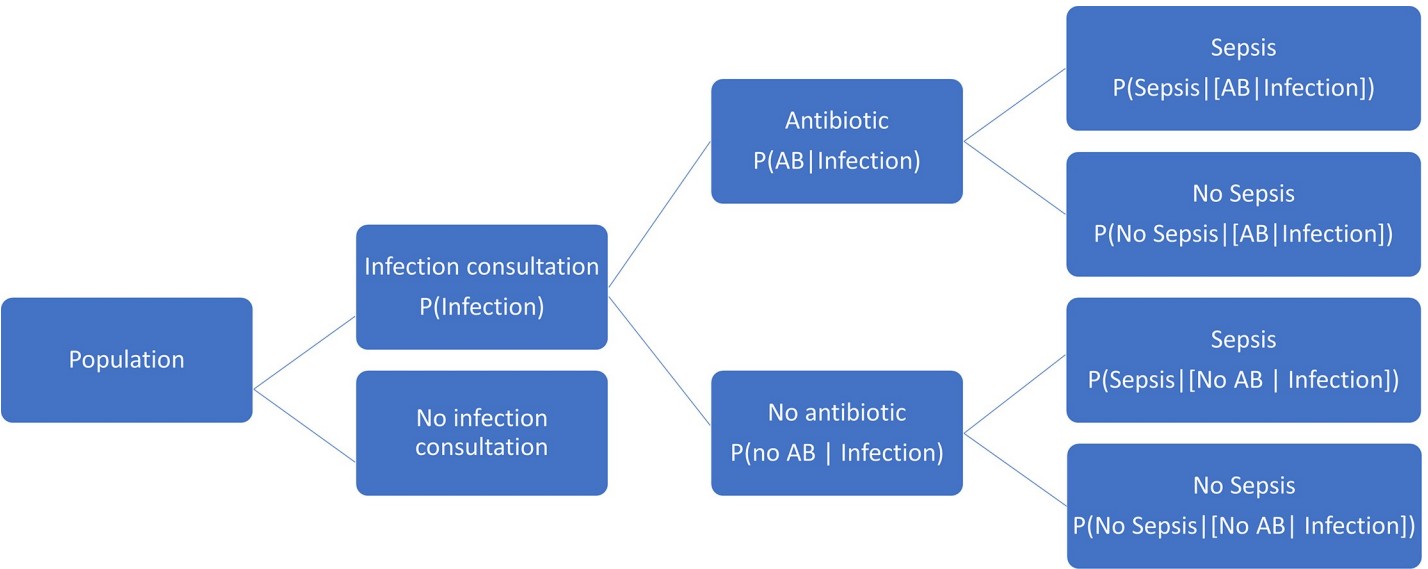

**Fig 1. Decision tree showing the probability of a patient consulting for an infection, being prescribed an antibiotic at that consultation, and developing sepsis.**
Please refer to Table 1 for explanation of abbreviations. AB, antibiotic; P, probability.

10-year age group. For the population aged 55 years and older, we also stratified by frailty category. We also evaluated subgroups of common infections, including RTI, skin infections, and UTI.

**Table 1. Definition and data source for probabilities.**

| Term | Explanation | Data source |
|---|---|---|
| P(Infection) | Probability of a person consulting with infection in a 30-day period | From infection consultation rate per 30 days in sampled data set from CPRD |
| P(AB \| Infection) | Probability of receiving an AB prescription on the same date as an infection consultation | From proportion of infection consultations with AB prescribed in sampled data set from CPRD |
| P(Sepsis) | Probability of sepsis, per 30 days | From incidence of sepsis from entire registered CPRD population |
| P(Infection \| Sepsis) | Probability of patients with sepsis having consulted for an infection in 30 days preceding their sepsis diagnosis | Proportion of sepsis cases with previous infection consultation, calculated from entire registered CPRD population |
| P(Sepsis \| Infection) | Probability of sepsis in the 30 days following an infection consultation | $\frac{P(\text{Infection}|\text{Sepsis})\ P(\text{Sepsis})}{P(\text{Infection})}$ |
| P(Sepsis \| [AB \| Infection]) | Probability of sepsis having consulted for an infection and received an AB prescription | $\frac{P([AB|\text{Infection}]|\text{Sepsis})P([\text{Sepsis}|\text{Infection}])}{P(AB|\text{Infection})}$ |
| P(Sepsis \| [NoAB \| Infection]) | Probability of sepsis having consulted for an infection and not received an AB prescription | $\frac{P([NoAB|\text{Infection}]|\text{Sepsis})P([\text{Sepsis}|\text{Infection}])}{P(NoAB|\text{Infection})}$ |
| NNT | The number of additional antibiotic prescriptions required to prevent one case of sepsis | $\frac{1}{P(\text{Sepsis}|[AB|\text{Infection}])-P(\text{Sepsis}|[No\ AB|\text{Infection}])}$ |

Abbreviations: AB, antibiotic; CPRD, Clinical Practice Research Datalink; NNT, number needed to treat; P, probability

## Sensitivity analyses

In sensitivity analyses, we evaluated whether use of a 60-day time window gave different results from a 30-day time window. The primary analysis reported data for a 16-year period, but the incidence of recorded sepsis has been increasing [24]. We repeated the analysis using only data for 4-year periods from 2002–2005 to 2014–2017 to evaluate whether estimates differed from the whole period from 2002 to 2017. We also investigated whether estimates differed if sepsis diagnoses recorded in Hospital Episode Statistics (HES) or as causes of death on mortality certificates were included. The sample for linkage was obtained from CPRD (Linkage Set 16). The linked sample included data for 378 English general practices, with 5,524,983 patients providing primary care electronic records data linked to HES and mortality statistics. We searched for ICD-10 codes for sepsis and septicaemia. We included primary diagnoses from HES-admitted patient care records and all mentions of sepsis in mortality statistics data. We repeated analyses using primary care electronic health records alone, primary care electronic health records with linked HES data, or primary care electronic health records with linked HES and mortality data.

## Results

The study included 706 general practices, with a total of 66.2 million person-years of follow-up (S1 Fig). Data for the distribution of sepsis patients by age and gender are shown in Table 2; data by region and period are shown in S3 Table. The probability of a consultation with a common infection of the skin, RTI, or UTI in any 30-day period ranged between 0.02 (1 in 50) and 0.08 (1 in 12). This probability of an infection consultation was higher in children and old

**Table 2. First sepsis events recorded in CPRD from 2002 to 2017 and preceding infection consultations and AB prescriptions.**

| Gender | Age group (years) | Sepsis events | Infection consultations in previous 30 days | Proportion (%) of sepsis events preceded by infection consultations | AB prescriptions on same date | Proportion (%) of infection consultations with ABs prescribed |
|---|---|---|---|---|---|---|
| Male | 0–4 | 224 | 51 | 22.8 | 11 | 21.6 |
| | 5–14 | 303 | 48 | 15.8 | 6 | 12.5 |
| | 15–24 | 360 | 59 | 16.4 | 21 | 35.6 |
| | 25–34 | 449 | 78 | 17.4 | 18 | 23.1 |
| | 35–44 | 791 | 117 | 14.8 | 24 | 20.5 |
| | 45–54 | 1,342 | 241 | 18.0 | 47 | 19.5 |
| | 55–64 | 2,466 | 472 | 19.1 | 102 | 21.6 |
| | 65–74 | 3,933 | 724 | 18.4 | 155 | 21.4 |
| | 75–84 | 4,752 | 1,089 | 22.9 | 256 | 23.5 |
| | 85+ | 2,738 | 713 | 26.0 | 158 | 22.2 |
| Female | 0–4 | 204 | 55 | 27.0 | 12 | 21.8 |
| | 5–14 | 238 | 32 | 13.4 | 9 | 28.1 |
| | 15–24 | 500 | 76 | 15.2 | 24 | 31.6 |
| | 25–34 | 806 | 110 | 13.6 | 38 | 34.5 |
| | 35–44 | 1,095 | 175 | 16.0 | 41 | 23.4 |
| | 45–54 | 1,631 | 267 | 16.4 | 72 | 27.0 |
| | 55–64 | 2,443 | 445 | 18.2 | 119 | 26.7 |
| | 65–74 | 3,215 | 646 | 20.1 | 180 | 27.9 |
| | 75–84 | 3,982 | 890 | 22.4 | 204 | 22.9 |
| | 85+ | 3,772 | 984 | 26.1 | 222 | 22.6 |

Abbreviations: AB, antibiotic; CPRD, Clinical Practice Research Datalink

people and greater in women than men during midlife (Tables 2 and 3). The probability of an antibiotic being prescribed at an infection consultation ranged between 0.43 and 0.67, with the probability being lowest for young children in whom consultation rates are highest (Table 3).

There were 35,244 first episodes of sepsis between 2002 and 2017. The probability of an infection consultation within 30 days before a sepsis event ranged between 0.14 (1 in 7) and 0.26 (1 in 4), with higher values at the extremes of age (Table 3). If no antibiotic was prescribed, the probability of sepsis at age 0–4 years was 0.000054 (1 in 18,519 consultations) in males and 0.000060 (1 in 16,667) in females. The probability of sepsis following an infection consultation without antibiotics increased linearly with age on a log scale (Fig 2, upper panel), reaching 0.004647 (1 in 215 consultations) in males and 0.003110 (1 in 321 consultations) in females aged 85 years and older (Table 3). If antibiotics were prescribed at an infection consultation, the estimated probability of sepsis was lower, ranging from 0.000020 (1 in 50,000 consultations) in males and 0.000023 (1 in 43,478 consultations) in females at age 0–4 years and to 0.000833 (1 in 1,200 consultations) in males and 0.000509 (1 in 1,965 consultations) in females aged 85 years and older. The number of antibiotic prescriptions required to prevent one sepsis event was highly age dependent (Fig 2, lower panel). For children aged 0–4 years, the NNT was 29,773 (18,458–71,091) in males and 27,014 (16,739–65,709) in females. However, at age 85 years and older, the NNT was 262 (236–293) in males and 385 (352–421) in females.

In the population aged 55 years and older, estimates were obtained separately by frailty category (Fig 3, S7 Table). The probability of sepsis was greater, and the NNT smaller, for patients with more advanced frailty. For nonfrail patients aged 65–74 years, the NNT was 1,680 (1,354–2,133) for men and 2,718 (2,089–3,697) for women. But for patients of the same age with severe frailty, the NNT was 259 (196 to 360) for men and 438 (329 to 624) for women. For patients with severe frailty, the NNT was less than 250 in men and less than 400 in women for all age groups over 55 years. For nonfrail patients, the probability of sepsis increased, and the NNT decreased, with increasing age (Fig 3). In nonfrail patients, the NNT declined from 2,309 (1,890–2,879) in men and 3,782 (3,001–4,907) in women at 55–64 years old to 407 (274–677) in men and 499 (346–780) for women at 85 years and older. Estimates for patients with mild or moderate frailty exhibited an intermediate pattern (Fig 3).

The probability of sepsis was higher following consultations for UTI than for skin infections or RTI, a pattern of association that was observed across all age groups and men and women (Fig 4, S8 Table). For patients aged 65 without antibiotic treatment, the probability of sepsis following an RTI consultation was 0.00090 (1 in 1,111 consultations) in men and 0.00053 (1 in 1,887 consultations) in women; following a skin infection consultation, the probability was 0.00224 (1 in 446) in men and 0.00150 (1 in 667) in women; following a UTI consultation, the probability was 0.009227 (1 in 108) in men and 0.003787 (1 in 264) in women. At the same age, the corresponding numbers needed to treat were as follows: for RTI, the NNT for men was 1,257 (1,112–1,434), and the NNT for women was 2,278 (1,965–2,686); for skin infection, the NNT for men was 502 (398–646), and the NNT for women was 784 (602–1,051); for UTI consultations, the NNT for men was 120 (102–145), and the NNT for women was 284 (241–342) (Fig 4).

## Sensitivity analyses

An analysis employing a 60-day time window to evaluate exposure gave generally similar results to those using a 30-day time window. In men aged 85 and over, the NNT for all infections was 262 (236–293) with a 30-day time window but 313 (276–359) with a 60-day window; for women of the same age, the figures were 385 (352–421) and 466 (419–523), respectively. When the analysis results were compared for the 4-year periods from 2002–2005 to 2014–

**Table 3. Probability of sepsis after infection consultations in primary care.**

| | | Probability of. . . | | | | | | |
|---|---|---|---|---|---|---|---|---|
| | | Infection consultation per 30 days | First sepsis event in any 30-day period | Infection consultation 30 days before sepsis event | AB at infection consultation | Sepsis after infection consultation, no AB | Sepsis after infection consultation, AB | |
| Gender | Age (years) | P(Infection) | P(Sepsis) | P(Infection \| Sepsis) | P(AB \| Infection) | P(Sepsis \| [No AB \| Infection]) | P(Sepsis \| [AB \| Infection]) | NNT (95% UI) |
| **Male** | 0–4 | 0.08 | 0.000014 | 0.23 | 0.43 | 0.000054 | 0.000020 | 29,773 (18,458–71,091) |
| | 5–14 | 0.04 | 0.000006 | 0.16 | 0.48 | 0.000047 | 0.000008 | 25,606 (17,962–40,817) |
| | 15–24 | 0.02 | 0.000008 | 0.17 | 0.58 | 0.000101 | 0.000041 | 16,921 (10,285–39,551) |
| | 25–34 | 0.02 | 0.000009 | 0.17 | 0.60 | 0.000193 | 0.000039 | 6,517 (4,779–9,522) |
| | 35–44 | 0.02 | 0.000013 | 0.15 | 0.62 | 0.000239 | 0.000039 | 5,035 (3,980–6,610) |
| | 45–54 | 0.02 | 0.000022 | 0.18 | 0.62 | 0.000472 | 0.000071 | 2,497 (2,121–2,999) |
| | 55–64 | 0.02 | 0.000048 | 0.19 | 0.63 | 0.000825 | 0.000135 | 1,449 (1,282–1,652) |
| | 65–74 | 0.03 | 0.000105 | 0.18 | 0.64 | 0.001305 | 0.000202 | 907 (823–1,007) |
| | 75–84 | 0.04 | 0.000219 | 0.23 | 0.63 | 0.002700 | 0.000478 | 450 (413–492) |
| | 85+ | 0.05 | 0.000416 | 0.26 | 0.61 | 0.004647 | 0.000833 | 262 (236–293) |
| **Female** | 0–4 | 0.08 | 0.000014 | 0.27 | 0.43 | 0.000060 | 0.000023 | 27,014 (16,739–65,709) |
| | 5–14 | 0.04 | 0.000005 | 0.14 | 0.51 | 0.000025 | 0.000010 | 65,522 (35,239–240,067) |
| | 15–24 | 0.04 | 0.000012 | 0.15 | 0.61 | 0.000080 | 0.000024 | 18,120 (12,472–30,241) |
| | 25–34 | 0.04 | 0.000016 | 0.14 | 0.63 | 0.000105 | 0.000033 | 13,926 (10,044–21,273) |
| | 35–44 | 0.04 | 0.000018 | 0.16 | 0.66 | 0.000184 | 0.000030 | 6,513 (5,349–8,194) |
| | 45–54 | 0.03 | 0.000028 | 0.16 | 0.66 | 0.000278 | 0.000054 | 4,463 (3,756–5,421) |
| | 55–64 | 0.04 | 0.000048 | 0.18 | 0.67 | 0.000490 | 0.000088 | 2,486 (2,179–2,876) |
| | 65–74 | 0.04 | 0.000080 | 0.20 | 0.67 | 0.000793 | 0.000151 | 1,557 (1,388–1,758) |

*(Continued)*

**Table 3.** (Continued)

| | | Probability of... | | | | | | |
|---|---|---|---|---|---|---|---|---|
| | | Infection consultation per 30 days | First sepsis event in any 30-day period | Infection consultation 30 days before sepsis event | AB at infection consultation | Sepsis after infection consultation, no AB | Sepsis after infection consultation, AB | |
| Gender | Age (years) | P(Infection) | P(Sepsis) | P(Infection \| Sepsis) | P(AB \| Infection) | P(Sepsis \| [No AB \| Infection]) | P(Sepsis \| [AB \| Infection]) | NNT (95% UI) |
| | 75–84 | 0.05 | 0.000138 | 0.22 | 0.66 | 0.001525 | 0.000231 | 773 (705–847) |
| | 85+ | 0.05 | 0.000271 | 0.26 | 0.64 | 0.003110 | 0.000509 | 385 (352–421) |

Abbreviations: AB, antibiotic; NNT, number needed to treat; P, probability; UI, uncertainty interval

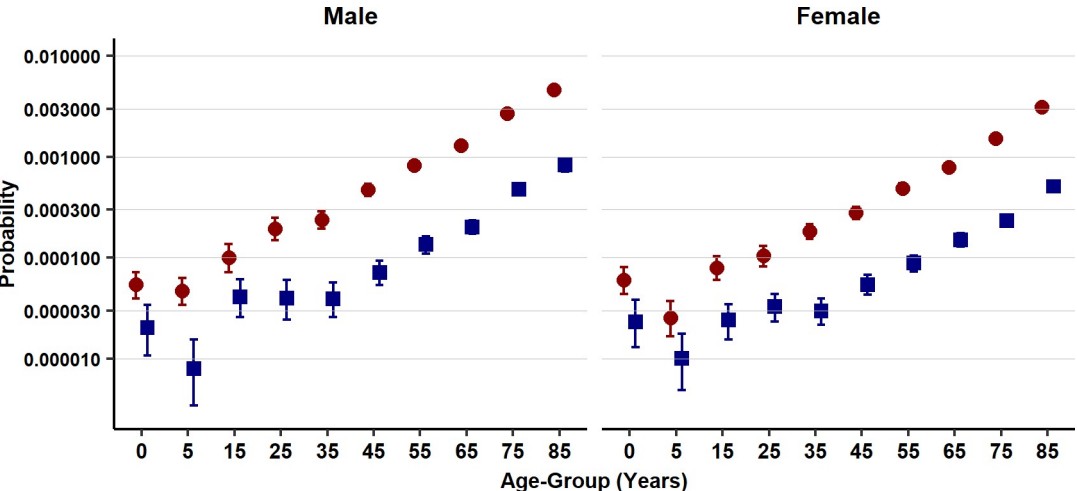

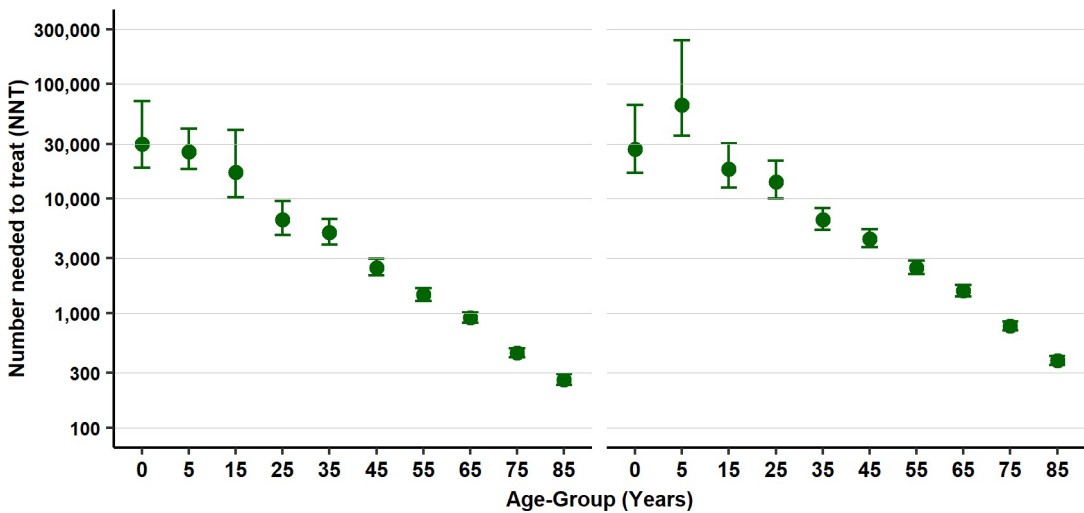

**Fig 2. Probability of sepsis following infection consultations in primary care if ABs are prescribed or not (upper panel).** Number of antibiotic prescriptions required to prevent one sepsis event (NNT) (lower panel). Figures are median probabilities (95% uncertainty interval). AB, antibiotic; NNT, number needed to treat.

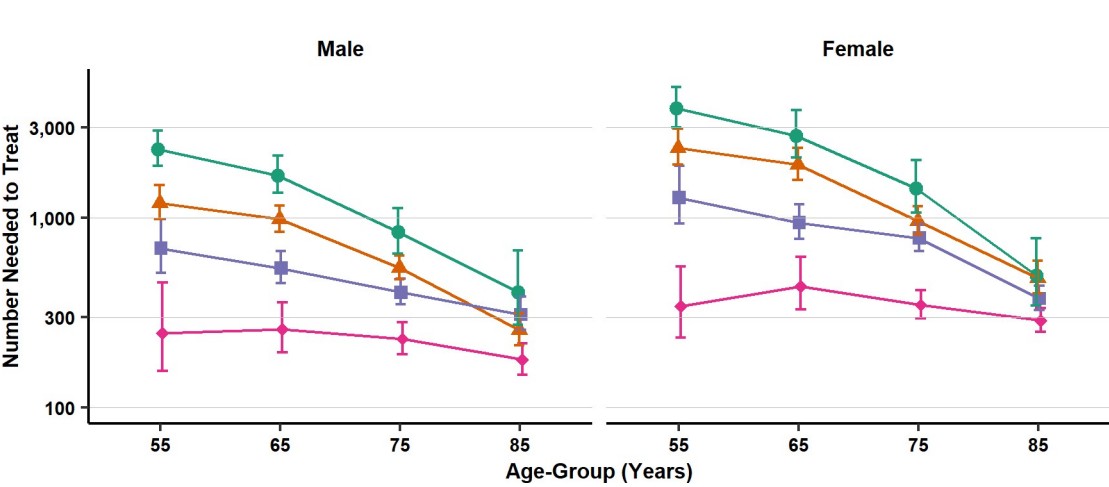

**Fig 3. Number of antibiotic prescriptions required to prevent one sepsis event (NNT) following infection consultations in primary care by frailty level, gender, and age group.** Figures are median estimates (95% uncertainty interval). NNT, number needed to treat.

2017, estimates for the probability of sepsis were slightly higher, and NNT slightly lower, for the most recent period (S2 Fig), consistent with the higher reported incidence of sepsis in this period (S9 Table). In the oldest age group, from 85 years and over, the probability of sepsis without antibiotics was as follows: for 2014–2017, the probability for men was 0.007287, and the probability for women was 0.004775; with antibiotics, the probability for men was 0.001290, and the probability for women was 0.000839; the NNT for men was 167 (141–202), and the NNT for women was 254 (216–302).

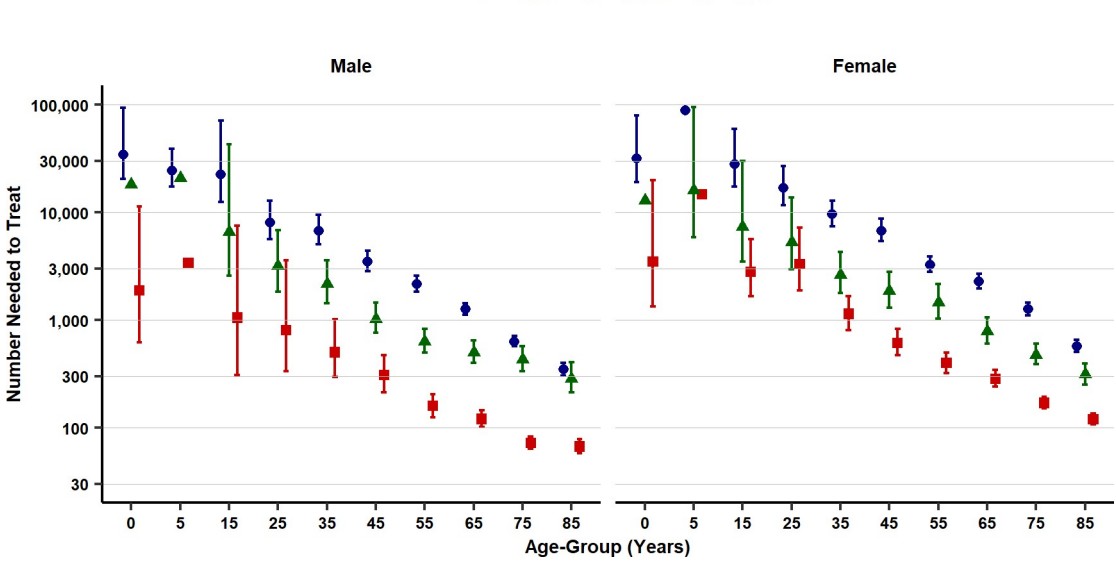

**Fig 4. Number of antibiotic prescriptions required to prevent one sepsis event (NNT) by age group, gender, and type of infection consultation.** Figures are median estimates (95% uncertainty interval). Uncertainty intervals were omitted for 0–4 years and 5–9 years if data were too sparse to give reliable estimates. NNT, number needed to treat; RTI, respiratory tract infection; UTI, urinary tract infection.

In the linked sample, there were 42,785 first sepsis events across all three data sources, including 17,341 from primary care records, 17,363 from HES admitted patient care (APC) primary diagnoses, and 8,081 from Office for National Statistics (ONS) mortality records during 36.2 million patient-years of follow-up. Accordingly, the underlying probability of sepsis was greater when linked records were employed. However, sepsis events in HES and ONS mortality statistics were less frequently associated with preceding infection consultations in general practice (S3 Fig). Consequently, the probability of sepsis following an infection consultation was only slightly higher if linked data were included in the analysis (S4 Fig), and the estimated NNT was only slightly lower (S5 Fig).

## Discussion

### Main findings

This study analysed primary care electronic health records data for a large population followed for 16 years with 35,244 new sepsis events. We found that the probability of sepsis following consultation for common infection episodes in primary care is highly age dependent. Without antibiotic treatment, sepsis may follow less than 1 in 10,000 infection consultations under 25 years of age and less than 1 in 1,000 consultations under 65 years of age. The probability of sepsis increases at older ages, and sepsis may follow approximately 1 in 200 (men) or 1 in 300 (women) consultations at age 85 or older. At older ages, the probability of sepsis is also highly dependent on frailty level: 55-year-olds with severe frailty have a similar probability of sepsis as a nonfrail 85-year-old. The probability of sepsis is related to infection type, with the greatest probability following consultations for UTI, the least for RTI, and consultations for skin infections being in an intermediate position. Risks were generally slightly higher for men, which might be accounted for by their generally lower consultation rates.

The incidence of recorded sepsis has been increasing over time with more inclusive case definitions and increasing awareness of the condition [24,30]. When we estimated the main results for the period from 2014 to 2017, the probability of sepsis was higher and NNT lower than for the period from 2002 to 2017. Although we caution that the absolute values of estimates may vary depending on the temporal or geographical context, we expect that in relative terms estimates will continue to identify older age, frailty, and UTI consultations as being associated with greatest risks of sepsis.

Sepsis is an uncommon but concerning outcome of common infection episodes in primary care. Appropriate antibiotic therapy may have immediate benefits that are not restricted to reduction in risk of sepsis, but antibiotic prescriptions are also often associated with immediate harms in the form of drug side effects. The potential risk of antimicrobial resistance has a significance that extends beyond the context of an individual consultation. Prescribing decisions must therefore be informed by the balance of all of the benefits and harms of either prescribing or not prescribing antibiotics. Quantification of the possible risks of sepsis contributes to informing these decisions.

### Strengths and limitations

The study drew on a large population-based cohort, enabling us to analyse representative data and obtain precise estimates that may be widely applicable. However, electronic health records comprise clinical data with several limitations and potential for bias. We analysed the outcomes of clinical decisions on whether to prescribe antibiotics or not. In the absence of randomisation, it may be expected that antibiotics were prescribed to individuals at higher risk, whereas lower-risk patients may be less likely to be prescribed antibiotics. Consequently, the probability of sepsis may be underestimated (in comparison with a study employing random

allocation) in the absence of antibiotics and overestimated for patients receiving antibiotics, with the NNT being overestimated. However, the analysis depended on general practice electronic health records of antibiotic prescriptions, which account for about 85% of community antibiotic prescribing [2], but we cannot exclude the possibility that patients might have obtained antibiotic prescriptions from alternative sources, including out-of-hours services. Measures of illness severity are rarely recorded for common infection consultations in primary care, so it was not possible to adjust for illness severity in analyses. It is also established that not all infection consultations in primary care are correctly coded, leading to underestimation of consultation rates [7]. We included data from 706 general practices over a 16-year period. Infection consultation and antibiotic prescribing rates were estimated from sample data. The estimates in this paper represent average values for this population of general practices and period of time. However, we conducted a sensitivity analysis with data from 2014 to 2017 only. We also acknowledge that in addition to changes in overall antibiotic utilisation, there have been changes in the proportion of prescriptions for broad-spectrum antibiotics. Future studies might be designed to compare the probability of sepsis if broad-spectrum or narrow-spectrum antibiotics are prescribed. The sample design used to estimate infection consultation rates and antibiotic prescribing proportions gave each practice, and each study year, equal weight, but we could have weighted the sample by practice size.

We analysed data for infection consultations in primary care and compared outcomes if antibiotics were or were not prescribed. However, previous studies showed that antibiotics may be prescribed at consultations with no definite diagnosis recorded [7,25]. We did not include these prescriptions because there was no valid comparator in terms of consultations without antibiotic prescriptions, but conclusions might have differed if missing diagnosis information had been complete. We caution that the precise values of these estimates may be expected to vary in different local contexts and according to the types of infection circulating in a community at a given time. We did not employ the approach of null hypothesis significance testing and do not report P values. We evaluated association modification by age, gender, frailty level, and consultation type. We employed the e-Frailty Index, which is a well-described measure based on 36 deficits [26], although we also applied it in the age range of 55–64 years, in which it is less well documented. We estimated stratified values for broad groups of patients, defined in terms of age, gender, and frailty. We did not estimate personalised risks for individual patients, and the clinical circumstances in each specific consultation should be used to inform estimates of sepsis risk for individuals. We relied on clinical records of sepsis from general practice, but we cannot be sure that all sepsis events were community rather than hospital acquired. In the UK, patients register with a family practice for continuing care, but patients may utilise emergency and out-of-hours services for acute problems such as sepsis, and these events might not be captured in primary care records. Providers may vary in their use of the term 'sepsis', which is an intermediate condition linking an infection and organ damage consequent on infection. The selection of clinical terms and medical codes is at the discretion of clinical staff, leading to lack of data standardisation. The conditions identified as 'sepsis' may represent a range of disease severity, and probability estimates might be proportionately lower if only severe sepsis was included. However, by using linked data, we showed that inclusion of hospital episodes and mortality records did not lead to any important changes in conclusions. Further research is needed to refine, update, and improve the accuracy of these initial estimates.

## Comparison with other studies

There has been a trend toward more-frequent recording of sepsis in recent years, and this has not always been accompanied by evidence of increased blood stream infections. In an

interrupted time series analysis, Balinskaite and colleagues [31] found no evidence that antimicrobial stewardship interventions in the UK might be associated with increased rates of sepsis. In an ecological analysis [24], we did not find evidence that general practices with lower antibiotic prescribing might have greater risk of sepsis over the same period of time and in the same practices as were included in the present study. Gharbi and colleagues [32] found that in older adults presenting with UTI, there was increased risk of sepsis if antibiotic prescriptions were not given or were delayed. The present results extend these findings by estimating risks across all ages, different levels of frailty, and different types of infection consultations. The lack of consistency between estimates from ecological- and individual-level analyses is likely to be explained by the substantial proportion of patients with sepsis who present without preceding infection consultations in primary care, as well as the small proportion of higher-risk consultations that are not associated with antibiotic prescriptions. RTI consultations are extremely frequent, which may account for the lower probability of associated sepsis. Respiratory infections are often the result of virus infections, and clinicians may tend to reserve the term 'sepsis' for bacterial infections. We evaluated uncomplicated lower UTIs, but estimates for the probability of sepsis might have been higher if kidney infections had been included.

## Conclusions

This paper quantifies the risk of sepsis following common infection consultations in primary care. These may be used in antimicrobial stewardship to identify groups of consultations at which reduction of antibiotic prescribing can be pursued more safely. The estimates show that risks of sepsis and benefits of antibiotics are generally more substantial among older adults, persons with more advanced frailty, or following UTI.

## Supporting information

**S1 STROBE Checklist. Items that should be included in reports of cohort studies.** STROBE, Strengthening the Reporting of Observational Studies in Epidemiology.
(DOC)

**S1 Table. List of Read codes for sepsis.**
(XLSX)

**S2 Table. List of Read codes for common infections.**
(XLSX)

**S3 Table. List of product codes for antibiotics.**
(XLSX)

**S4 Table. Proportion of consultations with antibiotics prescribed and consultation rates per person-year for different common infections.**
(DOCX)

**S5 Table. Estimated distribution of CPRD GOLD population by frailty level.** CPRD, Clinical Practice Research Datalink; PY, sum of person-years from 2002 to 2017.
(DOCX)

**S6 Table. Distribution of sepsis cases by gender, region, and period.**
(DOCX)

**S7 Table. Estimates by frailty category.**
(DOCX)

**S8 Table. Estimates by type of infection consultation.**
(DOCX)

**S9 Table. Sensitivity analysis using data for 2014–2017 only.** Column headings as main text
Table 2.
(DOCX)

**S1 Fig. Flow chart showing participant selection for main and linked samples.**
(DOCX)

**S2 Fig. Estimates for number of antibiotic prescriptions needed to prevent one sepsis episode (NNT) for four periods: 2002–2005 (blue), 2006–2009 (green), 2010–2013 (orange), and 2014–2017 (red).** NNT, number needed to treat.
(DOCX)

**S3 Fig. Probability of an infection consultation in primary care in the 30 days preceding a sepsis diagnosis using CPRD (linked sample) records (red); CPRD and linked HES records (blue); or CPRD, HES, and linked ONS mortality records (green).** CPRD, Clinical Practice Research Datalink; HES, Hospital Episode Statistics; ONS, Office for National Statistics.
(DOCX)

**S4 Fig. Estimated probability (95% uncertainty interval) of a first sepsis event within 30 days of an infection consultation in primary care if antibiotics were prescribed.** CPRD (linked sample) records only (red); CPRD and linked HES records (blue); or CPRD, HES, and linked ONS mortality records (green). CPRD, Clinical Practice Research Datalink; HES, Hospital Episode Statistics; ONS, Office for National Statistics.
(DOCX)

**S5 Fig. Estimated number of antibiotic prescriptions (95% uncertainty interval) to prevent a first sepsis event within 30 days of an infection consultation in primary care.** CPRD (linked sample) records only (red); CPRD and linked HES records (blue); or CPRD, HES, and linked ONS mortality records (green). CPRD, Clinical Practice Research Datalink; HES, Hospital Episode Statistics; ONS, Office for National Statistics.
(DOCX)

## Acknowledgments

The SafeABStudy Group also includes Dr Olga Boiko, Dr Caroline Burgess, Dr Vasa Curcin, and Dr James Shearer.

The views expressed are those of the authors and not necessarily those of the NHS, the NIHR, or the Department of Health. The authors had full access to all the data in the study, and all authors shared final responsibility for the decision to submit for publication.

## Author Contributions

**Conceptualization:** Martin C. Gulliford, Joanne R. Winter, Emma Rezel-Potts, Catey Bunce, Robin Fox, Paul Little, Alastair D. Hay, Michael V. Moore, Mark Ashworth.

**Data curation:** Joanne R. Winter.

**Formal analysis:** Martin C. Gulliford, Judith Charlton, Joanne R. Winter, Xiaohui Sun, Emma Rezel-Potts.

**Funding acquisition:** Martin C. Gulliford, Paul Little, Alastair D. Hay, Mark Ashworth.

**Investigation:** Judith Charlton.

**Methodology:** Judith Charlton, Joanne R. Winter, Xiaohui Sun, Emma Rezel-Potts, Catey Bunce, Robin Fox, Michael V. Moore.

**Project administration:** Martin C. Gulliford, Mark Ashworth.

**Software:** Judith Charlton, Joanne R. Winter.

**Supervision:** Martin C. Gulliford, Robin Fox, Paul Little, Alastair D. Hay, Michael V. Moore, Mark Ashworth.

**Writing – original draft:** Martin C. Gulliford.

**Writing – review & editing:** Judith Charlton, Joanne R. Winter, Xiaohui Sun, Emma Rezel-Potts, Catey Bunce, Robin Fox, Paul Little, Alastair D. Hay, Michael V. Moore, Mark Ashworth.

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
