## [Editor Report · Decision Letter 0]

14 Apr 2020

Dear Dr Gulliford, 

Thank you for submitting your manuscript entitled "PROBABILITY OF SEPSIS AFTER INFECTION CONSULTATIONS IN PRIMARY CARE. Population based cohort study and decision analytic model" for consideration by PLOS Medicine.

Your manuscript has now been evaluated by the PLOS Medicine editorial staff and I am writing to let you know that we would like to send your submission out for external peer review.

Kind regards,

Artur Arikainen,

Associate Editor

PLOS Medicine

---

## [Decision Letter · Decision Letter 1]

30 Apr 2020

Dear Dr. Gulliford,

Thank you very much for submitting your manuscript "PROBABILITY OF SEPSIS AFTER INFECTION CONSULTATIONS IN PRIMARY CARE. Population based cohort study and decision analytic model" (PMEDICINE-D-20-01208R1) for consideration at PLOS Medicine. 

[LINK]

In light of these reviews, I am afraid that we will not be able to accept the manuscript for publication in the journal in its current form, but we would like to consider a revised version that addresses the reviewers' and editors' comments. Obviously we cannot make any decision about publication until we have seen the revised manuscript and your response, and we plan to seek re-review by one or more of the reviewers. 

We expect to receive your revised manuscript by May 21 2020 11:59PM. Please email us (plosmedicine@plos.org) if you have any questions or concerns.

We look forward to receiving your revised manuscript. 

Sincerely,

Artur Arikainen, 

Associate Editor 

PLOS Medicine

plosmedicine.org

1. We ask that you address all points raised by the 3 reviewers, but want to draw particular attention to the following: Please address the reviewers’ requests regarding better reporting of your methodology. In particular, we ask that you provide a detailed response to the comments of reviewer #3 on the suitability of the dataset for your analyses. Please also consider the comments of reviewer #2 on whether it would be possible to include analyses by antibiotic class, though we recognise that this may not be feasible at this stage.

2. Please revise your title according to PLOS Medicine's style, using sentence case and a colon before the study type. Please also mention that the study is based on data from the UK.

3. Please structure your abstract using the PLOS Medicine headings (Background, Methods and Findings, Conclusions). Please combine the Design, Setting, Participants etc. sections into one section: “Methods and findings”. In the last sentence of the Methods and Findings section, please describe the main limitation(s) of the study's methodology. Please also include summary demographic information (eg. age, sex) for the cohort as a whole.

4. Please use the term ‘patient’ rather than ‘participant’ throughout your manuscript.

5. The Data Availability Statement (DAS) requires revision. For each data source used in your study: If the data are owned by a third party but available upon request, please note this and state the owner of the data set and contact information for data requests (web or email address). Note that a study author cannot be the contact person for the data. If the data are not freely available, please describe briefly the ethical, legal, or contractual restriction that prevents you from sharing it.

7. Please insert your citations before punctuation marks, eg: “…is attracting the concern of national governments and international organisations [1].” Please also ensure that your reference list is presented in the Vancouver style, including bold and italic formatting, where appropriate.

8. Please include a full list of the read codes used in your study, as Supporting Information.

9. Please provide line numbers throughout your manuscript text.

10. In the Abstract and Main Text, please include significance p values for the numerical data presented.

11. Please provide a table showing the baseline characteristics of the study population.

12. Please ensure that the study is reported according to the STROBE guidelines (https://www.strobe-statement.org/index.php?id=available-checklists), and include the completed STROBE checklist as Supporting Information. When completing the checklist, please use section and paragraph numbers, rather than page numbers. Please add the following statement, or similar, to the Methods: "This study is reported as per the Strengthening the Reporting of Observational Studies in Epidemiology (STROBE) guideline (S1 Checklist).”

13. Regarding reference [28] listed as ‘in press’, papers cannot be listed in the reference list until they have been accepted for publication or are otherwise publicly accessible (for example, in a preprint archive). The information may be cited in the text as a personal communication with the author if the author provides written permission to be named. Alternatively, please provide a different appropriate reference. 

14. Where possible, please provide additional geographic information relating to your dataset: for instance, are all practices from a small number of cities or is it an even distribution through the UK?

15. You can omit the Data, Funding, Conflict of Interest, and Author Contributions information on page 14, as this is extracted from the data provided in the submission form.

----

Comments from the reviewers:

Reviewer #1: "PROBABILITY OF SEPSIS AFTER INFECTION CONSULTATIONS IN PRIMARY CARE. Population based cohort study and decision analytic model" attempts to estimate the contribution of antibiotic prescription towards the prevention of sepsis, on over 66 million person-years of data collected from 706 U.K. general practices, from 2002 to 2017.

The major motivating concern is a trade-off between overprescription of antibiotics (wastage of resources, possible development of resistance by bacteria) and underprescription (reduces immediate patient safety, by increasing risk of follow-up bacteria infections, e.g. sepsis). This work focuses on the latter, and reports the estimated number of antibiotic prescriptions required to prevent one episode of sepsis (NNT) by age stratification. The extremely low incidence of sepsis in the month following an infection consultation (<0.0005, for all age groups) implies the need for an appropriately large set of data for correlations to be reliably drawn, a requirement that seems to be fulfilled. These NNTs were computed following a decision tree model, based on Bayes' theorem (Figure 1).

It was observed that NNT is always positive (i.e. antibiotic prescription is always correlated with reduced sepsis), and that NNT generally decreases with age (i.e. the contribution of antibiotic prescription towards reducing sepsis, increases with age). The relatively large values for NNT (minimum 262, maximum 65522) appear to mostly be a consequence of the extremely low incidence of follow-up sepsis, i.e. P(Sepsis|Infection), to begin with. The major limitation of this model, that the selection of patients receiving antibiotics is likely not random (i.e. P(AB|Infection) being likely dependant on other non-considered factors), was discussed; in any case, this likely leans towards NNT being overestimated, i.e. the impact of antibiotics on sepsis prevention being understated, from the presented results.

A particular strength of this study is its employment of a diverse set of records collected from hundreds of practices, covering tens of millions of patient-years, which allowed for a retrospective analysis of the antibiotics-sepsis relationship that would otherwise have been impractical through trials. The main decision tree model is clearly structured and described, appropriate sensitivity analyses were considered, and the conclusions appear broadly valid and of interest.

There do remain a number of possible clarifications:

1. In the "Sepsis events" section (Page 5), it is stated that "...we identified consultations for respiratory tract infections, skin infections and urinary tract infections because these are the most important groups of conditions for which antibiotics are prescribed in primary care". This however seems to imply that sepsis consultations for which antibiotic prescriptions were prescribed in the past month (i.e. for infections that weren't respiratory tract/skin/urinary tract), might have been ignored. While it is stated that these are the "most important group" of conditions for which antibiotics are prescribed, it is not clear whether they are also the most frequent group. In other words, was there a significant quantity of antibiotics prescriptions that were not covered under these three infection categories, and thus not analyzed? The supplementary data of the cited reference [25] was examined, but did not appear to cover this detail either.

2. In the "Selection of sample for antibiotic prescribing analysis" section, it is stated that a random sample of patients was drawn by selecting 10 participants for each gender/age group, also stratified by family practice.

a) From our understanding, the model probabilities presented in Table 1 were obtained from this set of sampled patients. If so, this might be explicitly stated.

b) It is not obvious as to why the stratification by practices was required. This seems to imply that, for example, a small practice with exactly ten female patients in the 0-4 year age group for some year would have all ten of them sampled for analysis, while a large practice with say 300 such patients would then also have just 10 of them sampled, with the remaining 290 ignored. If this description is correct, while this might reduce certain geographic-based biases, it would nevertheless seem to omit much entirely-valid data from consideration. Moreover, even if normalization by individual family practices is desired, another option might be to compute the required probabilities with all valid patients at the practice level, and then aggregate these probabilities equally, rather than sample (and discard data from consideration) early on.

c) Following from the above, it might be informative to have a flowchart collating the state of the data pre- and post-sampling (e.g. from the pre-sampled 66.2 million person-years to how many person-years post-sampling, from how many individual participants pre-sampling to 671,830 participants post-sampling, etc)

3. The degree of comprehensiveness of the EHR data might be discussed, i.e. might an individual patient visit one family practice for an infection, then visit a different practice/hospital upon onset of sepsis (not necessarily through referral)? If this is possible, would such cases be recognized/considered in the records/analysis? Also, given the relatively high mortality for sepsis (59,000 deaths from somewhat over 200,000 hospital admissions, from the Introduction), would deaths from sepsis (and possibly other complications) be expected to be recognized in the relevant records?

4. It is not discussed as to why infection/antibiotics prescription within the preceding 30 days was determined as the relavant period, rather than e.g. 60 days. Is 30 days a standard assumption for duration of antibiotic effect?

Reviewer #2: The authors sampled the CPRD GOLD primary care electronic health record data base to estimate the probability of consultation for an infection of the skin, respiratory tract or urinary tract, and used a decision tree approach to estimate the probability of sepsis following these consultations by whether an antibiotic was prescribed. Strengths of the study include population estimates by age, sex and frailty of this risk overall and by each infection, and provide estimates of the number of antibiotic prescriptions required to prevent one sepsis event for each of the above mentioned categories. The study uses a strong observational design and sampling method.

There are a number of areas where the presentation and interpretation of might strengthened. First, the outcome was determined by ICD codes (presumably assigned by treating physicians or administrative staff), so there is no uniform definition of the different categories. This should be acknowledged in the limitations. Second, sepsis, urosepsis and septicemia are considered to be equivalent and are lumped together in the presentation. These conditions may vary significantly in severity, and if sensitive to the prescribed antibiotic treatment can be straightforward (septicemia and urosepsis). At a minimum, it would be very helpful to provide separate estimates for sepsis. 

The two most common sources for sepsis are the lungs (pneumonia) and the kidneys. However, respiratory illnesses (including pneumonia) are often caused by viruses, although the viral infection may lead to secondary bacterial infection (e.g. influenza). Therefore, it is not surprising that the NNT is highest for respiratory infections. If there were a way to disentangle this group (perhaps presenting results for pneumonia, alone), would be useful. Similarly, it would be useful to separate UTI into cystitis and pyelonephritis.

Discussing how the authors believe their data might help antibiotics might be used more safely in greater detail would strengthen the manuscript. In this regard, it is unfortunate that the authors did not examine classes of antibiotics prescribed. One strategy for reducing emergence of antibiotic resistance is to minimize use of broad-spectrum antibiotics where possible. With additional analyses, the authors might speak more directly to this issue.

The justification for choosing sepsis as an outcome might be made more explicitly in the introduction and explored more thoroughly in the discussion. Appropriate antibiotic therapy has an immediate benefit that is not tied to the risk of sepsis. Sadly, the only way to limit the emergence of antibiotic resistance - which threatened our ability to treat severe infection such as sepsis - is to limit antibiotic use. It may not be reasonable to treat 30,000 individuals to prevent one case of sepsis, but there may be other reasons to do so. On the flip side - depending on the antibiotic - treating 30,000 individuals might results in 300 adverse side effects. 

Reviewer #3: The findings of this study could have important clinical implications for practitioners that fear missing sepsis. Expressing findings in NNT is very helpful and will support understanding the clinical relevance of the findings for frontline practitioners. 

My major concern with this study concerns the reliability of these estimates given they have drawn upon data from such a large period of time. The researchers are / were up against at least three major challenges: 1. Sepsis is a tricky subject for carrying out data-linkage studies because definitions have changed multiple times over the last couple of years alone, the awareness amongst both clinicians and the public has shifted dramatically, and there have been several preventive health interventions aiming to tackle delays in the identification and management of sepsis. 2. Depending on the license agreement with CPRD, there can be limits on the number of records extracted. 3. Best practice, particularly concerning antibiotic stewardship, has changed considerably since 2002 (18 years ago). To mitigate the influence of these challenges, I am perplexed they did not endeavour to sample more patients (to the maximum of their license) from more recent years (even 2012-2017), or at the very least in the manuscript explored this possibility +/- considered the implications of not doing so (they might have, and not had sufficient word count to do so here). The horse has bolted now and they can't fix this.

[LINK]

---

## [Decision Letter · Decision Letter 2]

28 May 2020

Dear Dr. Gulliford,

Thank you very much for re-submitting your manuscript "Probability of sepsis after infection consultations in primary care in the United Kingdom: population-based cohort study and decision analytic model" (PMEDICINE-D-20-01208R2) for review by PLOS Medicine.

I have discussed the paper with my colleagues and the academic editor and it was also seen again by two reviewers. I am pleased to say that provided the remaining editorial and production issues are dealt with we are planning to accept the paper for publication in the journal.

[LINK]

We look forward to receiving the revised manuscript by Jun 04 2020 11:59PM. 

Sincerely,

Artur Arikainen, 

Associate Editor 

PLOS Medicine

plosmedicine.org

Requests from Editors:

1. Please update the title to include the study dates: “Probability of sepsis after infection consultations in primary care in the United Kingdom in 2002-17: population-based cohort study and decision analytic model”

2. Please update your Competing Interests statement on the submission form to the following standard text: “The authors have declared that no competing interests exist.”

3. Please move the “Data sources” section from page 18 to either the Data Availability Statement in the submission form, or the Methods section of the main text, or remove it altogether.

4. In the Abstract, please include an additional limitation, eg. the possibility of missing or incorrect health record data, or possible sources of antibiotics outside primary care.

5. Please remove the keywords from page 2. Our published articles are indexed automatically using a controlled taxonomy.

6. Author summary: Please spell out UTI and RTI, for clarity to non-scientist readers.

7. Please include line numbers in your manuscript margin.

8. In the section “Data source”, please provide a URL link to the database website.

9. Please cite the study protocol the same way as with other references, rather than as a hyperlink, or include the URL in brackets.

10. In the section “Selection of sample for antibiotic prescribing analysis”, please include a brief description of how the random sample was chosen, eg. by computer-generated list.

11. There are some instances in the results where UTI and RTI are spelled out, even though the abbreviations are already used in earlier parts of the text, eg. page 12.

12. In the Discussion, please break up the long paragraph on limitations, in order to improve readability.

13. Thank you for addressing our comment relating to p values. Our only request is that you remove this sentence: “Readers may reflect on the substantive importance of estimated differences, and associated uncertainty intervals, for their work.”

14. Please correct this sentence in the Discussion to: “Future studies might be designed to compare the probability of sepsis if broad-spectrum or narrow-spectrum antibiotics are prescribed.”

15. Please format your references to strict Vancouver style – bold and italics are not used.

16. Please correct the typo in reference 9: “Antimicrobial”

17. Please provide more access details (eg. a URL) for references 17, 19, and please update reference 28 to include full details rather than “in press”. 

18. In the Discussion please replace ‘significant’ in the following sentence with a more appropriate term, eg. ‘notable’: “The lack of consistency between estimates from ecological- and individual-level analyses are likely to be explained by the significant proportion of patients…”

19. Please avoid the use of ‘effect’ throughout your text, given the observational nature of your study, eg. as in this sentence: “Age, gender and frailty were evaluated as effect modifiers.”

20. The terms gender and sex are not interchangeable (as discussed in http://www.who.int/gender/whatisgender/en/); please use the appropriate term.

21. Thank you for responding to our comment 14 in the previous decision letter. To clarify, where possible, we would like you to provide a summary of sepsis events broken down by region or NHS trust, eg. as Supporting Information.

-------

Comments from Reviewers:

Reviewer #1: We thank the authors for considering our previous suggestions. For Supplementary Figure 1, the arrows might be labelled with brief descriptions of the selection process for convenience, or the sampling description summarized as a caption. On the additional sensitivity analyses for 2002-2005 & 2014-2017, the authors might consider including the intervening four year periods (2006-2009, 2010-2013) as well for completeness, if it is not too much trouble.

Reviewer #3: My comments have been adequately investigated and now addressed in the manuscript with appropriate discussion of their implications. The sensitivity analyses highlight the fragility of these data when different assumptions are taken. The authors have sufficiently described such limitations in the main paper, and should consider reflecting this more explicitly in the abstract.

[LINK]

---

## [Editor Report · Decision Letter 3]

25 Jun 2020

Dear Prof. Gulliford, 

On behalf of my colleagues and the academic editor, Dr. Andrew Carson-Stevens, I am delighted to inform you that your manuscript entitled "Probability of sepsis after infection consultations in primary care in the United Kingdom in 2002-17: population-based cohort study and decision analytic model" (PMEDICINE-D-20-01208R3) has been accepted for publication in PLOS Medicine. 

PRODUCTION PROCESS

PRESS

PROFILE INFORMATION

Thank you again for submitting the manuscript to PLOS Medicine. We look forward to publishing it. 

Best wishes, 

Artur Arikainen, 

Associate Editor 

PLOS Medicine

plosmedicine.org